# Experimental Characterization of a Foil Journal Bearing Structure with an Anti-Friction Polymer Coating

**Grzegorz Żywica** [1,*], **Paweł Bagiński** [1], **Jakub Roemer** [2], **Paweł Zdziebko** [2], **Adam Martowicz** [2] and **Tomasz Zygmunt Kaczmarczyk** [1]

1 Department of Turbine Dynamics and Diagnostics, Institute of Fluid-Flow Machinery, Polish Academy of Sciences, 80-231 Gdansk, Poland
2 Department of Robotics and Mechatronics, AGH University of Science and Technology, 30-059 Krakow, Poland
* Correspondence: gzywica@imp.gda.pl

**Abstract:** The development of highly efficient and environmentally friendly machines requires the use of new technologies that are created using innovative design solutions and new materials. This also applies to various types of propulsion units, such as gas microturbines or combustion engines. Although these machines have been known for many years, by using new components, it is still possible to improve their performance. This article presents an experimental study conducted on a gas foil bearing using a polymer coating as an anti-friction material. These types of bearings allow for a reduction in friction losses and are not lubricated with conventional lubricants. The dynamic characteristics of the foil bearing structure were determined, which are essential in terms of both rotor dynamics and the entire propulsion system. The research was carried out over a wide range of frequencies, with different loads acting in different directions. Hysteresis loops and vibration orbits were determined. The authors showed that displacements perpendicular to the load in some cases may be relatively large and should not be ignored. The results obtained during the tests can be used to validate numerical models of such bearings, optimize their design and select the structural and anti-friction materials.

**Keywords:** foil bearing; journal bearing; oil-free bearings; polymer coating; dynamic characteristic

## 1. Introduction

Bearings are one of the key components of any machine with rotating parts. In general, they determine the durability and reliability of the machine in terms of its limitations, such as its maximum speed and load capacity. In this context, it can be stated that bearings have a crucial impact on the allowable operating range of the entire machine. It also follows that bearings are one of the most heavily loaded components, which means that much attention has to be paid to their analysis at all stages of the machine's life.

Currently, machine designers have a very wide range of options to select bearings for operating parameters, resulting from the specifics of the technological process and environmental conditions. The selection of commercially available, off-the-shelf bearings narrows as speed and temperature requirements increase. Currently, the most commonly used high-speed bearings are gas bearings [1], magnetic bearings [2,3] and high-precision rolling bearings [4,5], whose rolling elements can be made of ceramic materials. Each of these bearing types has some unique properties that make them preferred for specific applications. Unfortunately, at extremely high speeds, exceeding 100 krpm, and at elevated temperatures, only few types of bearings can operate reliably and ensure the long-term transmission of rotational motion and loads acting on the shaft. For these applications, gas foil bearings complement the group of gas bearings. Although these types of bearings have been known for several decades, it was only in the last two decades that they have seen their heyday, driven by demand resulting from the rapid development of various types

of high-speed machines such as microturbines [6–8]. That is why a lot of research and development work is being carried out to use foil bearings in the aircraft industry [9,10], distributed power generation [11–13] or the automotive industry, where attempts have been made to use them to support the rotor of a turbocharger [14–16]. Properly matched to the operating conditions, foil bearings can have many advantages, such as the absence of oil lubrication, low friction losses, reduced vibration level, high durability, and the possibility to increase the speed and operating temperature [17]. These advantages are a direct result of the design and operating principle of foil bearings.

The most characteristic component of a foil bearing consists of a set of thin foils that create a compliant structure with a variable geometry (Figure 1). At the moment when the shape of the top foil (which is supported in numerous places by bump foils) is adjusted to the current operating point of the journal, the load from the shaft is transferred to a larger area than in the case of rigid support. When the bearing is operating and the journal rotates at a sufficiently high speed, it is lifted by a gaseous layer, which is formed as a result of the hydrodynamic effect. Thus, above a certain speed, which is called 'lift-off speed', the journal is separated from the top foil. Under such working conditions, no wear occurs. Since, below the lift-off speed, the journal comes into contact with the top foil, it is necessary to apply appropriate protective coatings that reduce friction and wear [18].

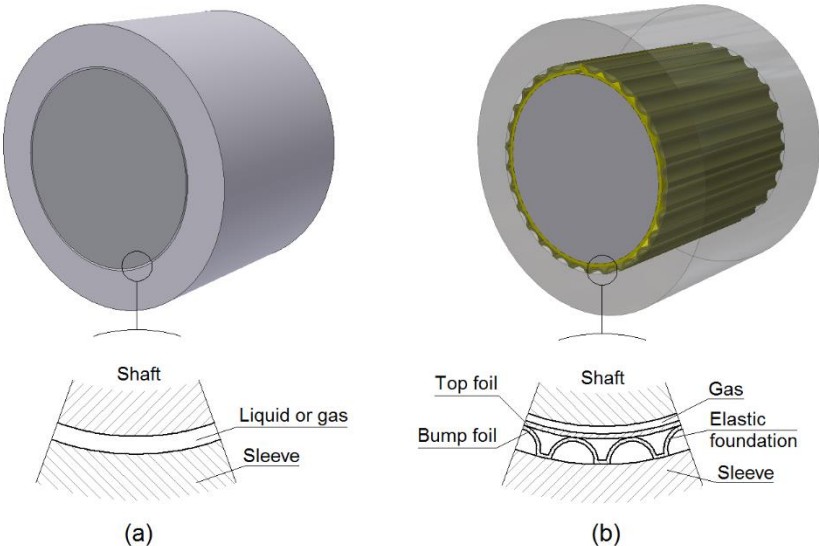

**Figure 1.** Comparison between a typical liquid/gas lubricated slide bearing (**a**) and a bump-type gas foil bearing (**b**).

Based on the literature review, it can be said that in low- and medium-temperature applications (up to 300 °C), thin polymer coatings are often used, which are made of polytetrafluoroethylene (PTFE), polyimide or polyamide [19,20]. Polymeric materials are also combined with solid lubricants (such as molybdenum disulphide, graphite and boron nitride) to reduce friction and improve durability [21–23]. The contact surface of the journal is usually coated with chromium (dense chromium or hard chromium). For higher temperatures (above 300 °C), foil coatings should be prepared with very hard, wear-resistant ceramics (such as titanium carbide and aluminium oxide) or metal–ceramic–solid lubricant composites [24]. At the highest operating temperatures, bare or hard-coated foils are used with ceramic or composite shaft coatings (such as plasma-sprayed PS-series coatings [25]). These materials provide an acceptable level of friction, and the foil bearings can operate at a temperature of 650 °C or higher [19,21]. Under certain operational conditions, the use of nanocomposites [26] or the texturing of sliding surfaces [27] can also produce beneficial effects.

Various modifications of foil bearings, in which the bump-type foil structure is replaced by different constructions that exhibit beneficial properties, are proposed. These

include metal mesh and hybrid bump-metal mesh foil bearings [28], gas foil bearings with metal shims between the bumps and the sleeve [29], foil bearings with an embedded compression spring [30] or smart materials [31]. Variations such as a hybrid (hydrodynamic and hydrostatic) gas foil bearing [32] and multi-layer protuberant foil bearings [33] are also under development. In some systems with foil bearings, various cooling techniques are used, which can be based on a cooling fan [34], side-feed pressurization [8] or thermocouples [35]. A state-of-the-art survey on the development of foil bearing technology can be found in several publications [6,36].

Modeling of the dynamic properties of the foil bearing structure, due to its mechanical complexity, still remains a huge challenge. Gas foil bearings operate with a relatively thin hydrodynamic lubricating film, the thickness of which usually does not exceed a dozen or so micrometres [37,38]. Consequently, the overall stiffness and damping of a gas foil bearing depends on both the gas film and the compliant supporting structure. The underlying compliant structure also determines the ultimate load capacity of a bearing [39]. Therefore, when analysing the properties of an entire foil bearing, besides a lubricant flow model, a structural model of the foils should be taken into account [40,41]. In a model that takes into account fluid–structure interactions, the foil assembly model can be treated as a submodel and should be included in rotordynamic analysis. Depending on the needs and operating conditions, different fluid–structure interactions and thermal phenomena can also be taken into account in a model of a complete foil bearing [42,43].

One of the aspects to take into account when analysing the phenomena that take place in the bearing structure is the friction processes that occur between the various components of the bearing [44]. In a mechanical system with multiple points of linear or surface contact between parts, friction has a significant effect on energy dissipation and vibration damping [45]. This is why advanced friction models, which, for example, take into account changes in the coefficient of friction versus the sliding speed, are increasingly used to describe these phenomena [46]. For most solid materials, the static friction coefficient is greater than the kinetic friction coefficient. In lubricated systems, the friction force normally decreases as the velocity increases from zero [47]. This phenomenon is called the Stribeck effect and is presented in Figure 2. Based on this figure, it can be said that when two surfaces are sliding across each other, the friction force varies, and the kinetic friction coefficient is not directly proportional to the sliding velocity. Therefore, the dynamic characteristics of the foil bearing structure can significantly change depending on the frequency and amplitude of the exciting force, which translate into the sliding velocity. To assess the structural forces acting in the foil bearing under certain conditions, the equivalent viscous damping and dry friction force are usually specified. The values of these parameters can be calculated based on the work performed during the motion. During an experimental test, it can be determined based on the work done by the vibration exciter using the equations given in many publications (e.g., [48–51]). Based on the equations given in these papers, the structural stiffness and equivalent viscous damping of the bearing structure can be determined.

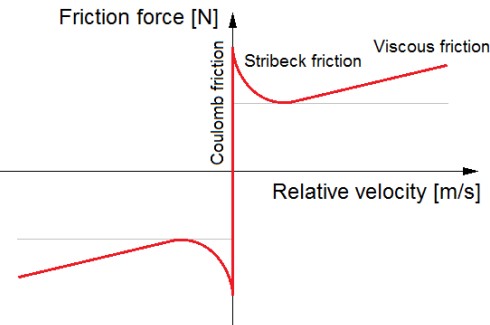

**Figure 2.** Graph illustrating the friction model based on the Stribeck effect with viscous friction.

This article discusses an experimental study aimed at determining the dynamic characteristics of a foil bearing structure, in which an anti-friction polymer coating was applied to the sliding surface of the top foil. Besides the hysteresis loops commonly used in these types of studies (which are created based on the course of the displacements of a component to which a certain force is applied), the trajectories of the movement in the plane perpendicular to the bearing axis were also determined and analysed. This required very accurate measurements of the displacements in the direction of the load and also in the perpendicular direction. The research thus conducted constitutes a novelty in terms of the study of the dynamic characteristics of a foil bearing structure, and the results obtained provide new insight into the dynamic processes that occur in such systems. This is because it has been demonstrated that under certain loading conditions, the displacements in the direction perpendicular to the direction of the excitation force can be relatively large or even larger than the displacements in the direction in which the load acts. In the approach used so far, which is widely presented in the literature, it has usually been assumed that the transverse displacements are small enough to be neglected when determining the stiffness and damping parameters of a foil bearing structure. Based on the study carried out, the authors showed that this assumption is an oversimplification, and that in some cases, the approach used so far needs to be modified. Based on the conducted research, the authors of this paper showed that due to the specific properties of the bump-type foil bearing, the transverse displacement cannot be ignored. This claim turned out to be true even when we take into account the dynamic properties of only the structural part of the foil bearing.

The following sections of the article discuss the most important issues regarding the research conducted. Section 2 discusses in detail the foil bearing studied and the test rig. The research methodology is also discussed in this section. Next, Section 3 presents the results of the study, together with their discussion. Several conclusions of a general nature are formulated at the end of the article.

## 2. Materials and Methods

### 2.1. Object of Investigation

The object of investigation is a gas foil bearing with a nominal diameter of 34 mm and a width of 40 mm (Figure 3a). The foil bearing has a single top foil and three bump foils spaced along its perimeter. All foils are fixed to the bearing sleeve using grooves (with angular increments of 120°). Each bump foil has seven bumps that are evenly spaced along the curvature of the bearing sleeve. The geometry of the bump foil was obtained using an original manufacturing method employing 3D printed tools [52]. Each bump foil is also divided into four sections across the width of the bearing. The top foil and the bump foils were made of a nickel-based superalloy—Inconel Alloy 625. The inner side of the top foil, which is in contact with the journal, has a wear-resistant anti-friction coating (Figure 3b) made of a specially selected polymer (polytetrafluoroethylene-based material). The selected coating material has the trade designation AS20 and is characterized by a low coefficient of friction (the static friction coefficient is 0.1, and the kinetic friction coefficient is 0.04 [46]), high abrasion resistance and a maximum operating temperature of 280 °C. The thickness of the protective coating on the top foil is between 10 to 15 μm. This coating is necessary to protect the top foil during run-ups and run-downs. The basic dimensions of the bearing are shown in Figure 4, and its main technical parameters are summarized in Table 1.

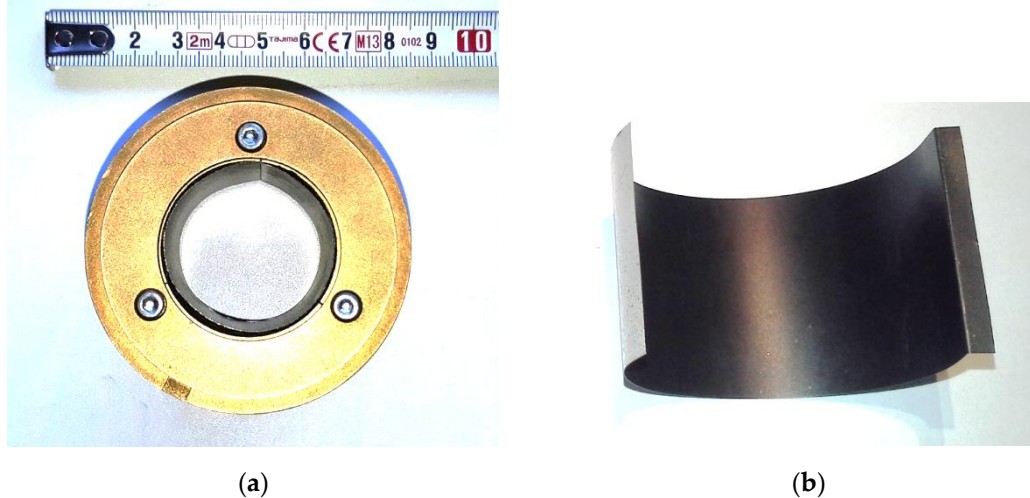

(**a**)                              (**b**)

**Figure 3.** Pictures of tested foil bearings: (**a**) Assembled bearing prepared for testing; (**b**) Top foil with anti-friction coating before assembly.

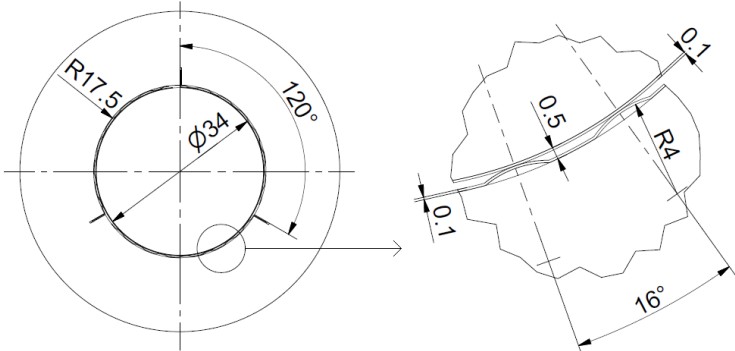

**Figure 4.** Basic dimensions of the tested foil bearing with a nominal diameter of 34 mm.

**Table 1.** Main technical data of the tested foil bearing.

| Parameter | Value |
|---|---|
| Journal diameter | 34 mm |
| Sleeve inner diameter | 35 mm |
| Bearing length | 40 mm |
| Bearing mass | 930 g |
| Number of top foils | 1 |
| Number of bump foils | 3 |
| Top foil thickness | 0.115 mm |
| Bump foil thickness | 0.1 mm |
| Number of bumps of a single bump foil | 7 |
| Bump radius (inner) | 4 mm |
| Bump height | 0.3 mm |
| Angle between bumps | 16° |
| Journal material | steel (1.7225) |
| Journal coating (0.25 mm thick) | $Cr_2O_3$ |
| Top/bump foil material | Inconel 625 |
| Top foil coating (10–15 μm thick) | AS20 (PTFE based) |
| Sleeve material | bronze (CuSn10P) |

The foil bearing has been designed for use in real rotating machines. The geometry of the bearing has been selected in such a way as to obtain a load capacity of 25 N and a minimum level of vibrations in a wide range of rotational speeds (up to 40 krpm). In order to

keep the top foils in the correct position during operation, two rings were attached on both sides of the bearing sleeve. The prototype foil bearing can be fitted with a thermocouple that measures the temperature of the top foil, but since the shaft was not rotating during the tests in question, the temperature sensor was removed. The bearing was prepared for the experimental tests. Its assembly preload was obtained by using a new set of foils (one top foil and three bump foils), and its surfaces (the top foil and journal) had to be run in. Detailed information on the preparation of the bearing for dynamic tests is given in Section 2.3.

### 2.2. Description of the Test Rig

The experimental tests were carried out using a test rig dedicated to determining the dynamic characteristics of a foil bearing structure. The bearing sleeve was placed inside a housing (with a mass of 950 g) which was connected via a stringer (with a diameter of 5.5 mm and a length of 160 mm) with an electromagnetic shaker. The shaft with the bearing journal was held in place by two supports, as shown in Figure 5. Bearing sleeve displacements were measured using a high-precision laser sensor (Keyence LK-H050, Osaka, Japan) with a measurement range of ±10 mm, linearity of ±0.02 and repeatability of 0.025 μm, which was connected to an LK-G5001 controller. The resolution of the displacement measurement was set to 0.1 μm. Dynamic excitation was generated by an electromagnetic vibration exciter (TIRA TV 51110, Schalkau, Germany) with a rated peek force of 100 N, a frequency range of 2–7000 Hz and a maximum peak-to-peak displacement of 13 mm. The excitation force was adjusted individually for each series of measurements using an amplifier (TIRA BAA-120, Schalkau, Germany), whose signal-to-noise ratio is greater than 95dB and distortion is greater than 0.05%, so as to obtain amplitude values on the desired level. During the experimental tests, the actual force generated by the shaker, which acted on the bearing, was measured using a mechanical impedance sensor with a quartz sensing element (PCB Piezotronics 288D01, Depew, NY, USA), with a measurement range of ±222.4 N and non-linearity of ±1%. The impedance sensor was mounted between the stringer and the bearing housing. A multichannel analyser (LMS SCADAS Mobile 05, Leuven, Belgium) with dedicated software (LMS Test. Lab 11B, Leuven, Belgium) was used as the data acquisition system. Figure 5 shows a picture of the test rig, taken during the foil bearing test.

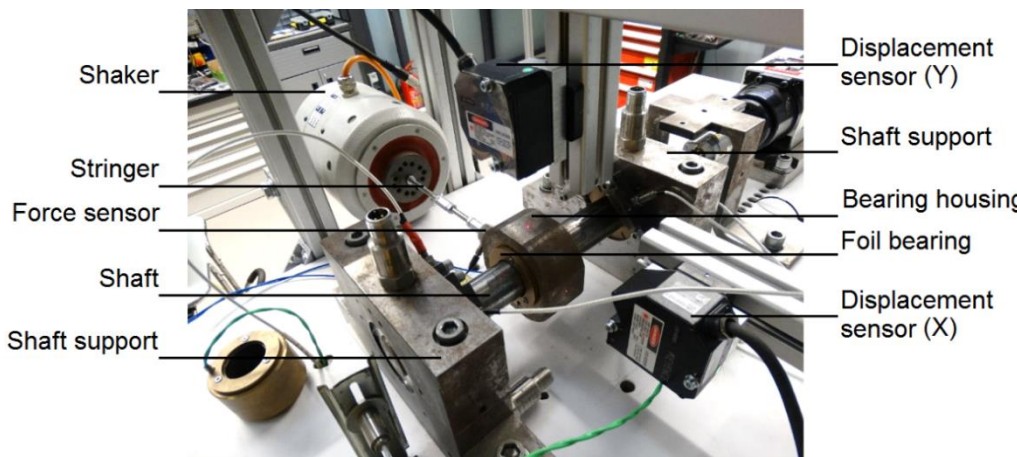

**Figure 5.** Picture of the foil bearing test rig during dynamic tests performed with a vibration exciter.

The test rig described above was originally designed to test rotating shafts supported by two foil bearings. Therefore, a drive system is visible in the upper right corner of Figure 5, but it was not used during the tests discussed in this article; the coupling connecting the drive to the shaft has been dismantled. The foil bearings, which had been placed inside two supports, were replaced by rigid sleeves, the diameters of which were appropriately

selected. In this way, the shaft, on which the tested bearing was mounted, was firmly attached to the main plate of the test rig via bearing supports. The main plate, together with the other components of the test rig, was placed on an individual foundation, isolated from ambient vibrations by means of anti-vibration mats. The vibration exciter was attached to a separate frame (placed on a separate foundation), so the vibrations it generated were only transmitted to the bearing through the stringer. This configuration of the test rig has ensured the high accuracy and repeatability of measurements.

### 2.3. Experimental Procedure

The following measurement procedure was used to determine the dynamic properties of the foil bearing structure. A new top foil and new bump foils were mounted in the test bearing. Prior to the dynamic tests, the bearing had to be run in on a separate test rig (designed for testing foil bearings with a floating sleeve). The bearing run-in process was carried out by gradually increasing the speed to 24 krpm and was completed when the top foil temperature was stable at this speed. The bearing thus prepared was placed in the housing mounted on the test rig used for the dynamic tests (Figure 5). During these tests, the bearing sleeve with the housing rested on the rigid shaft, and was supported only by compliant foils (i.e., a single top foil and three bump foils). The sleeve vibrations in the horizontal direction were excited using a harmonic force, the frequency and amplitude of which were precisely adjusted. To evaluate the properties of the bearing structure in two mutually perpendicular directions, the sleeve in which the foils are fixed was rotated by 90° in the bearing housing. Thanks to this, in the first configuration, the attachment point of the top foil was at the top (the groove of the sleeve was at right angles to the direction of the acting load), and in the second configuration, the attachment point of the top foil was aligned with the direction of the acting load. The two bearing configurations analysed are presented in Figure 6.

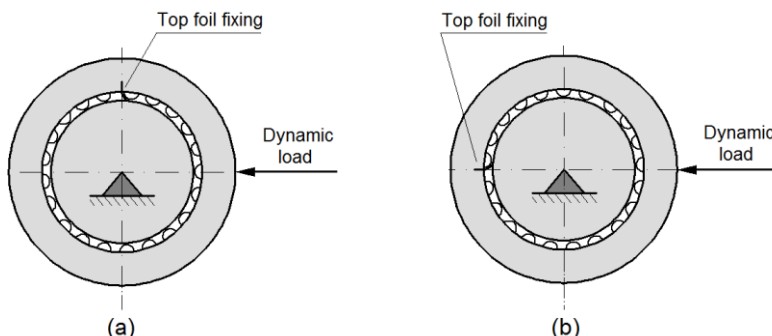

**Figure 6.** Two foil bearing configurations used in the research: (**a**) Load acting in the direction perpendicular to the sleeve groove where the top foil was fixed; (**b**) Load acting in the direction parallel to the sleeve groove.

Bearing tests with two orientations relative to the direction of excitation were carried out using five frequencies (i.e., 20 Hz, 40 Hz, 80 Hz, 150 Hz and 300 Hz) and two amplitudes of the harmonic force (25 N and 50 N). The frequency and amplitude values were chosen based on the preliminary tests carried out on this bearing and the experience gained to date by the authors of the article, so that the most significant characteristics of the tested bearing could be captured using a limited number of measurement series. This approach allowed for capturing changes in the characteristics resulting, for example, from the passage of the system through resonance, and for limiting the number of graphs at which the vibrations are negligible. When selecting the amplitude of the harmonic force, the parameters of the vibration exciter used and the bearing capacity were taken into account. The maximum load value of 50 N corresponds to the maximum force amplitude that we were able to obtain in a long-term test over the entire frequency range. The value of 25 N corresponds to the nominal load capacity of the tested foil bearing. With suitably chosen experimental

conditions, even if the number of series of measurements and the amount of recorded data are not large, the influence of the foil bearing structure on the dynamic performance of the rotor supported by such bearings can be accurately assessed. In addition, this study aimed at acquiring experimental data to validate the numerical models of foil bearings and making these data available to others. A summary of all measurement series, in which different frequencies and amplitudes of the applied force, as well as two bearing configurations, were taken into account, is presented in Table 2. The research carried out covered a total of 20 measurement campaigns.

**Table 2.** Main parameters used in all series of measurements.

| Frequency [Hz] | Configuration (a) | | Configuration (b) | |
|---|---|---|---|---|
| | Force [N] | | | |
| 20 | 25 | 50 | 25 | 50 |
| 40 | 25 | 50 | 25 | 50 |
| 80 | 25 | 50 | 25 | 50 |
| 150 | 25 | 50 | 25 | 50 |
| 300 | 25 | 50 | 25 | 50 |

During the tests, the excitation frequencies were gradually increased from the lowest value to the highest value. For each excitation frequency, the tests were started at a force amplitude of 25 N, and after performing the measurements, the load was increased until it reached an amplitude of 50 N, and then the measurements were performed again. After performing the measurements at the highest frequency, the vibration exciter was turned off, and the bearing configuration was changed, as shown in Figure 6. Then, all measurements were repeated starting with the lowest frequency and ending with the highest frequency, taking into account two load levels at each frequency. At each frequency and load, the tests lasted a few minutes, and the proper measurement was only performed after the system vibrations had stabilized at a constant level (it is important to note that the values recorded by all the sensors were displayed in real time on the screen of the measurement system). Only then did the recording of the signals from the sensors begin, which lasted 0.5 s. The sampling frequency of the recorded data was 25,600 Hz, which allowed the force and displacement waveforms to be accurately represented even at the highest excitation frequency. In addition, the signals from the displacement sensors were subjected to band-pass filtering, during which all components differing by ±1 Hz from the current excitation frequency were cut off. All graphs presented in the following section of the article have been drawn up based on the results obtained during the last full load cycle, which was recorded by the data acquisition system.

## 3. Results and Discussion

This section of the article presents the results of an experimental study conducted on a foil bearing structure that was loaded using a vibration exciter. The amplitude and frequency of the dynamic force were precisely controlled using an amplifier, and the sleeve displacements were measured with very high accuracy using optical sensors. The measurements thus performed were used to draw up two types of characteristics. In Section 3.1, diagrams showing the displacement of the sleeve versus the current magnitude of the dynamic force (so-called hysteresis loops) are presented. Section 3.2 provides additional characteristics that show the vibration trajectories of the sleeve depending on the direction of the load as well as the amplitude and frequency of the excitation force. In order to facilitate the analysis of the results, the maximum values of displacement on the axes of the graphs had been chosen individually for each excitation frequency, while they remained the same for the two load directions presented side by side. As for the trajectories of the sleeve displacements (Section 3.2), the same scale was used on the ordinate and abscissa axes. This allows for an easier interpretation of the results.

### 3.1. Hysteresis Loops

Hysteresis loops have a very wide range of applications. In the case of dynamic studies of a foil bearing structure, they are used to illustrate the course of displacements in the same direction as the direction of excitation depending on the current value of the harmonic force. First, they can be used to show the amplitude of vibration versus the amplitude, frequency and direction of the dynamic load. Second, they can be used to evaluate the damping properties of the bearing structure, since the area of a hysteresis loop is proportional to the dissipated energy [46]. Based on the shape of a hysteresis loop (or, more precisely, the angle of the major axis of the resulting ellipse with respect to the axis of the coordinate system), it is also possible to classify whether the vibration of the system takes place before, during or after the resonance. Based on these characteristics, the stiffness and damping of the dynamic system can also be determined [48,49].

Figures 7–11 below show the displacement characteristics versus the dynamic load of the foil bearing structure, which were determined experimentally for the bearing discussed in Section 2. Since studies conducted by many researchers have shown that the direction of the load, depending on the position of the top foil mounting location, can have a significant effect on the dynamic characteristics of the foil bearing [53], tests were conducted with two mutually perpendicular load directions, where the load acted perpendicularly on and parallel to the top foil mounting groove in the sleeve (as shown in Figure 6). As the foil bearing structure is a nonlinear system, the tests were carried out using two excitation forces of different magnitude, namely, 25 N and 50 N. The excitation frequencies were chosen by the authors of the article based on their previous experiments conducted on similar foil bearings [46,54], and these frequencies help capture the most significant dynamic phenomena that occur in the system under study, without having to present many very similar characteristics.

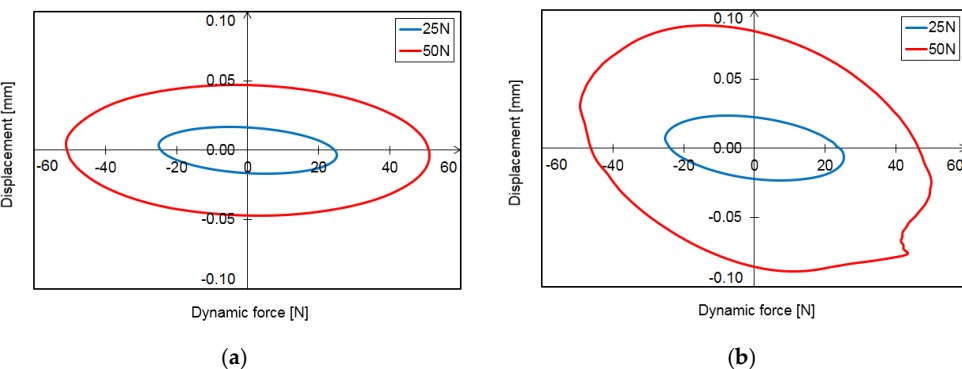

(**a**)             (**b**)

**Figure 7.** Experimental displacements of the foil bearing sleeve versus dynamic force at a frequency of 20 Hz: (**a**) Load acting in the direction perpendicular to the sleeve groove; (**b**) Load acting in the direction parallel to the sleeve groove.

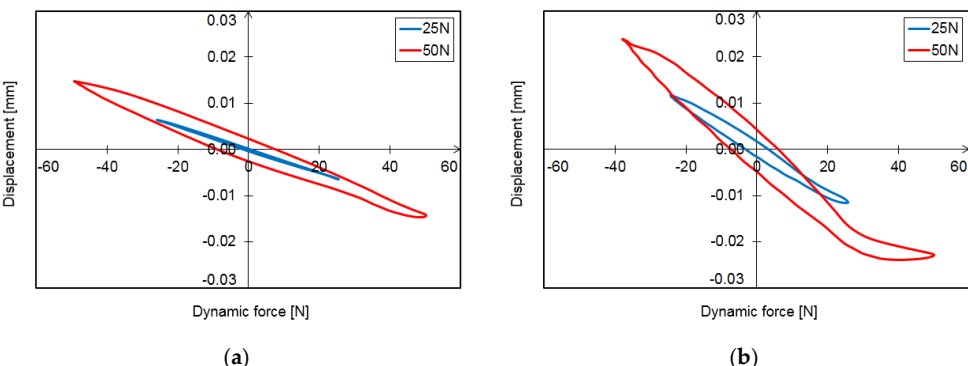

(**a**)             (**b**)

**Figure 8.** Experimental displacements of the foil bearing sleeve versus the dynamic force at a frequency of 40 Hz: (**a**) Load acting in the direction perpendicular to the sleeve groove; (**b**) Load acting in the direction parallel to the sleeve groove.

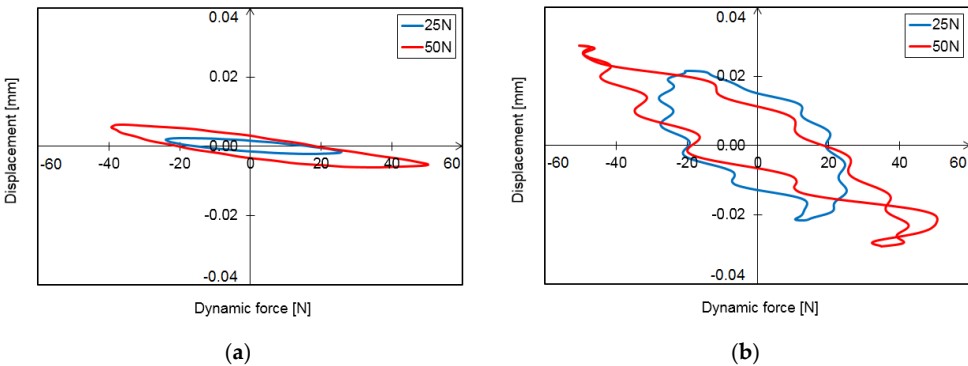

**Figure 9.** Experimental displacements of the foil bearing sleeve versus dynamic force at a frequency of 80 Hz: (**a**) Load acting in the direction perpendicular to the sleeve groove; (**b**) Load acting in the direction parallel to the sleeve groove.

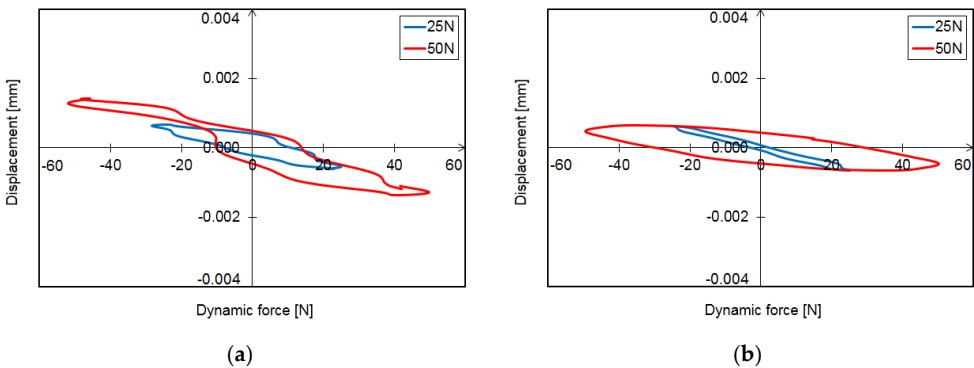

**Figure 10.** Experimental displacements of the foil bearing sleeve versus the dynamic force at a frequency of 150 Hz: (**a**) Load acting in the direction perpendicular to the sleeve groove; (**b**) Load acting in the direction parallel to the sleeve groove.

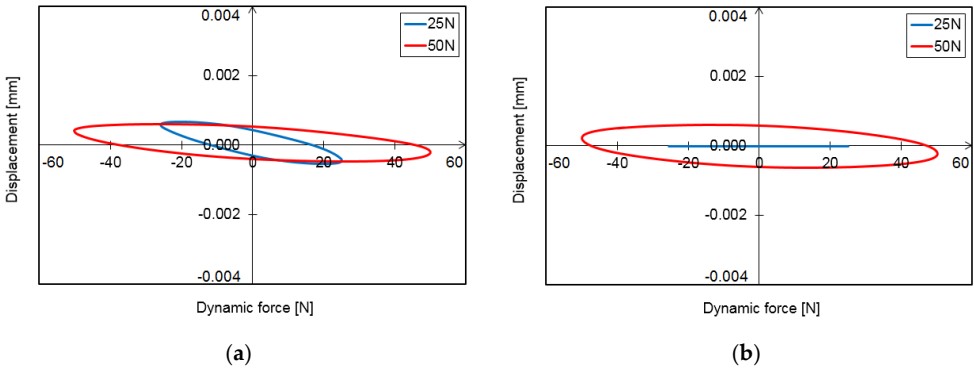

**Figure 11.** Experimental displacements of the foil bearing sleeve versus the dynamic force at a frequency of 300 Hz: (**a**) Load acting in the direction perpendicular to the sleeve groove; (**b**) Load acting in the direction parallel to the sleeve groove.

Figure 7 shows the results of the study conducted on the foil bearing structure at a frequency of 20 Hz. The characteristics obtained for the first load variant (Figure 7a) show very regular loops of an elliptical shape, the axes of which correspond to the axes of the coordinate system. A twofold increase in excitation force amplitude (from 25 N to 50 N) resulted in an almost threefold (2.81) increase in displacement (from 0.0168 mm to 0.0472 mm) in the same direction as that of the acting load. The shape of the loop, however, had not changed. The 90° change in the direction of the load caused some changes in the dynamic characteristics (Figure 7b). At a load of 25 N, the maximum displacement

increased by 40% (from 0.0168 mm to 0.0236 mm), and at a load of 50 N, by 88% (from 0.0472 mm to 0.0889 mm). Besides the increase in vibration amplitude, the shape of the resulting loop also changed, becoming irregular in the fourth quadrant of the coordinate system. In this configuration of the test stand, the load through the foils acted on the attachment point of the top foil (according to Figure 6b). Under a heavy load, at the level of 50N, there could be a slight movement of the end of the top foil, which was fixed in the groove of the sleeve. Some structural clearance in the groove where the foil was attached, which appeared at the frequency of 20 Hz, could have influence on the fluctuation of the measured force and the displacement.

Figure 8 shows the hysteresis loops obtained at an excitation frequency of 40 Hz. Even at first glance, it is clear that the characteristics have changed significantly compared to the results obtained at a frequency of 20 Hz. The major axes of the resulting loops pass through the second and fourth quadrants of the coordinate system, and although the range of displacements on the abscissa axis has been significantly reduced (from 0.1 mm to 0.03 mm), the loops are significantly flattened. At a load of 25 N, the maximum displacement for each loading method was 0.0064 mm (Figure 8a) and 0.0116 mm (Figure 8b), respectively, representing an 81% increase. At a load of 50 N, the displacements were 0.0147 mm (Figure 8a) and 0.0240 mm (Figure 7b), respectively. Therefore, the percentage increase was 63%. Besides the change in vibration amplitude, some changes in the shape of the loops were also observed.

Figure 9 shows the displacements of the foil bearing sleeve, which were obtained using a dynamic excitation frequency of 80 Hz. As for the results obtained for the first loading method (Figure 9a), loops similar to those at a frequency of 40 Hz were obtained (Figure 8a), except that the major axes of the loops approached the horizontal axis of the coordinate system. The increase in load resulted in an increase in maximum displacement from 0.0023 mm to 0.0062 mm (a 170% increase). Tests of the second method of loading the system at a frequency of 80 Hz showed a significant change in the characteristics of the bearing (Figure 9b). Although the major axes of the resulting loops still passed through the second and fourth quadrants of the coordinate system, the loop itself had a very irregular shape. This could be due to the stick-slip phenomenon in this system, which interfered with the smooth movement of the mating parts and caused pulsations in the values of force recorded by the impedance sensor. As the load increased from 25 N to 50 N, the system vibrated more, and the maximum displacements changed from 0.0217 mm to 0.0289 mm; therefore, the percentage increase was 33%. Comparing the results obtained using different bearing load directions, it can be concluded that when the load acted in parallel to the sleeve notch (Figure 9b), the vibration amplitude increased by nine times at 25 N, and almost by five times at 50 N, compared to the variant where the direction of the load was perpendicular to the sleeve notch (Figure 9a).

With a further increase in the excitation frequency, a decrease in the system's vibration amplitude was observed. This can be seen in Figures 10 and 11, where the displacement scale has been increased tenfold (the maximum value on the vertical axis has been reduced from 0.04 mm to 0.004 mm). It is important to note that the quantitative analysis of the results presented in Figures 10 and 11 may lead to erroneous conclusions since the recorded amplitudes of the sleeve vibrations were of the order of 0.001 mm, which was close to the resolution of the measurement paths with the optical displacement sensors used. In addition, the fact that the vibration of the mechanical system at the level of 1–2 μm is of little practical significance should be mentioned. Therefore, the results obtained at the frequencies of 150 Hz and 300 Hz are discussed without referring to exact values.

In the case of the results presented in Figure 10, it can be seen that an increase in load from 25 N to 50 N resulted in an increase in the vibration level only in the case of the first method of positioning the bearing in relation to the excitation force (Figure 10a). In the second load variant (Figure 10b), the vibration amplitude remained almost unchanged. A similar phenomenon can be observed when the foil bearing structure is subjected to a force with a frequency of 300 Hz, the direction of which is perpendicular to the groove

in the sleeve (Figure 11a). Despite a twofold increase in the amplitude of the excitation force, the sleeve displacements still remained at the same level. When the bearing was rotated by 90° (Figure 11b), at a higher load (50 N), the results were almost identical to those obtained before the bearing was rotated, but at a lower load (25 N), the measurement system recorded almost no movement of the system under study. When the vibration disappeared, it was deemed unnecessary to continue testing at higher frequencies since the test object behaved like an infinitely rigid system.

*3.2. Vibration Trajectories*

The hysteresis loops shown in the previous section are typical dynamic characteristics of the foil bearing structure. Although this is a very useful form to present the measurement results, it does not give a complete picture of the phenomena that occur in such a system when a harmonic force acts on it. However, it is important to note that in such mechanical systems, a force acting in a particular direction causes the loaded part of the system to move not only in the same direction but also in a direction perpendicular to the direction of the acting force. This is due to the design of the foil bearing structure and the interactions that occur between the bearing parts. These interactions are often defined by so-called cross-coupled stiffness and damping coefficients, which, unlike direct coefficients, are determined by taking into account the displacement that takes place in the direction perpendicular to the acting force. The movement of the loaded part of the foil bearing (i.e., movement of the journal or sleeve) in the direction perpendicular to the direction of the acting load occurs both during test rig tests using a vibration exciter and during the operation of such a bearing in a rotating machine. When a load acts in a specific direction on a rotating shaft that is supported by foil bearings, displacement also occurs in a direction perpendicular to the direction of the acting load. In this case, this is not only due to the properties of the gas in the lubrication gap, but rather to the characteristics of the structural support layer Itself.

In order to present a complete picture of the movement of the foil bearing sleeve relative to the stationary journal, during the tests that were conducted, the displacements of the sleeve were measured in two directions using high-precision displacement sensors (as shown in Figure 5). Such test results perfectly complement the previously presented hysteresis loops and have not yet been published in the literature. According to the authors of this article, the presentation of such results provides very valuable information on the dynamic properties of a foil bearing structure and can help explain the phenomena that occur there. It also allows numerical models to be fully verified without having to make the questionable assumption that a force acting on such a system will cause displacements only in the same direction in which it acts.

Based on the test results shown in Figures 12–16, it is clear that the displacements in the direction perpendicular to the direction of excitation were significant enough that they should certainly not be ignored. In some cases, the displacements in this direction were even greater than those in the direction of the acting load. Figure 12 shows the vibration orbits of the foil bearing sleeve obtained under dynamic excitation at a frequency of 20 Hz. In the first configuration (Figure 12a) with a force of 25 N, the displacement of the sleeve in the horizontal direction was smaller than that in the vertical direction (0.0168 mm and 0.0238 mm, respectively). With a force of 50 N, the opposite situation occurred, since the maximum displacement in the horizontal direction was 0.0472 mm, and in the vertical direction, 0.0352 mm. When the bearing structure was loaded after it was rotated by 90° (Figure 12b), the displacements in the horizontal direction were greater than in the vertical direction regardless of the magnitude of the acting force (0.0236 mm and 0.0153 mm, respectively, for 25 N; 0.0889 mm and 0.0124 mm, respectively, for 50 N). In all the cases shown in Figure 12, the vibration orbits of the sleeve have regular elliptical shapes, but the proportions of the loops obtained and their orientation with respect to the coordinate system vary. It is clearly visible that it depends not only on the position of the attachment point of the foil relative to the direction of the force, but also on the magnitude of the force.

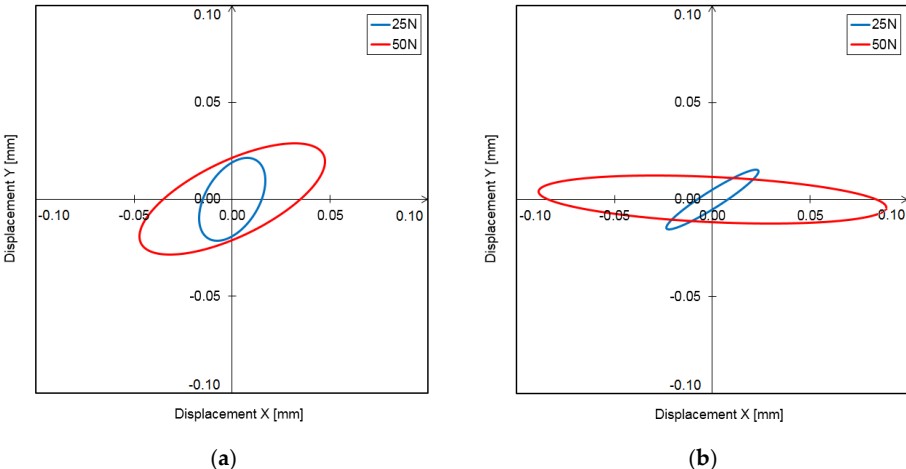

**Figure 12.** Experimental displacements of the foil bearing sleeve in two directions under a dynamic load at a frequency of 20 Hz: (**a**) Load acting in the direction perpendicular to the sleeve groove; (**b**) Load acting in the direction parallel to the sleeve groove.

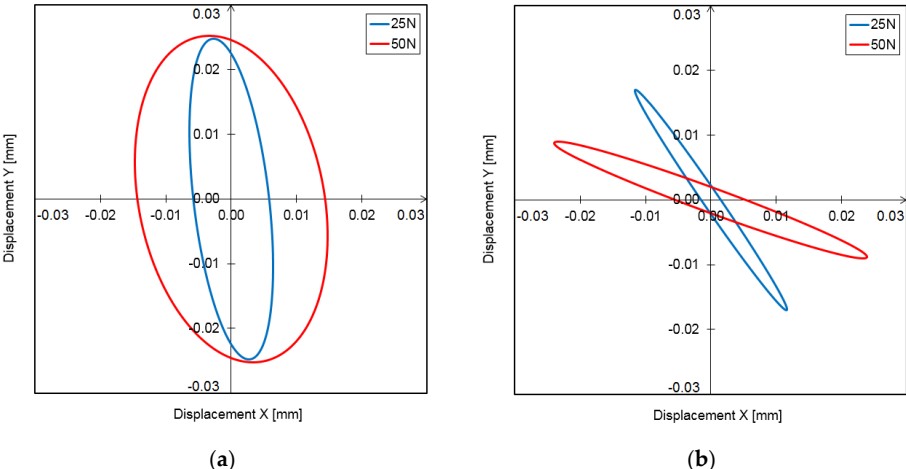

**Figure 13.** Experimental displacements of the foil bearing sleeve in two directions under a dynamic load at a frequency of 40 Hz: (**a**) Load acting in the direction perpendicular to the sleeve groove; (**b**) Load acting in the direction parallel to the sleeve groove.

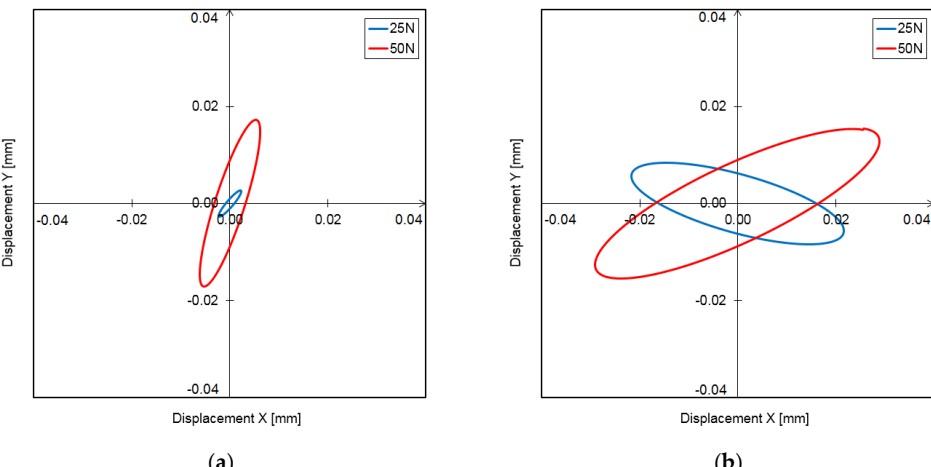

**Figure 14.** Experimental displacements of the foil bearing sleeve in two directions under a dynamic load at a frequency of 80 Hz: (**a**) Load acting in the direction perpendicular to the sleeve groove; (**b**) Load acting in the direction parallel to the sleeve groove.

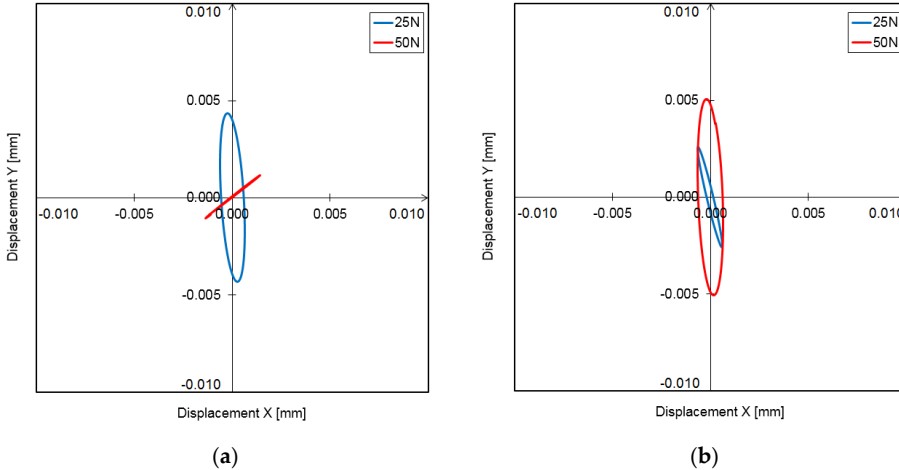

(**a**)  (**b**)

**Figure 15.** Experimental displacements of the foil bearing sleeve in two directions under a dynamic load at a frequency of 150 Hz: (**a**) Load acting in the direction perpendicular to the sleeve groove; (**b**) Load acting in the direction parallel to the sleeve groove.

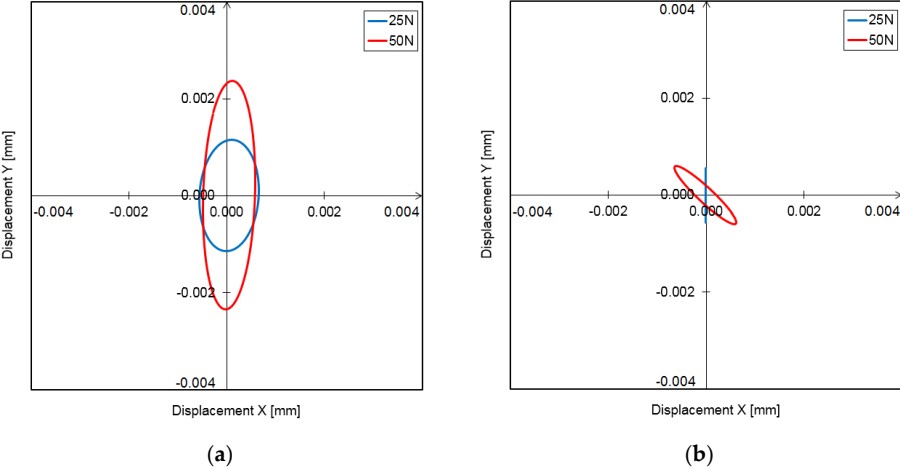

(**a**)  (**b**)

**Figure 16.** Experimental displacements of the foil bearing sleeve in two directions under a dynamic load at a frequency of 300 Hz: (**a**) Load acting in the direction perpendicular to the sleeve groove; (**b**) Load acting in the direction parallel to the sleeve groove.

Figure 13 shows the results obtained using an excitation frequency of 40 Hz. In Figure 13a, it can be seen that in two load cases, the maximum displacements in the *X* direction were smaller than those in the *Y* direction (0.0064 mm and 0.0245 mm, respectively, for 25 N; 0.0147 mm and 0.0252 mm, respectively, for 50 N). Interestingly, in both cases, the displacement in the direction perpendicular to the direction of the acting force was very similar, about 0.025 mm. As for the load variant whose results are shown in Figure 13b, it can be observed that the obtained loops have a similar shape; however, the major axis of the loop obtained with a higher load is more inclined towards the *X*-axis. The maximum displacements from the equilibrium position in the directions of the *X* and *Y* axes are as follows: 0.0116 mm and 0.0169 mm, respectively, for 25 N; 0.0240 mm and 0.0089 mm, respectively, for 50 N. This example clearly demonstrates that the position where the foil is mounted relative to the direction of the acting load has a very strong influence on the vibration trajectories.

It Is therefore clear that the following characteristics, shown in Figure 14, confirm the above statement. A twofold increase in excitation frequency caused a considerable change in the shapes of the obtained loops, and when the bearing sleeve was rotated by 90°, there were also, in this case, large differences between the results shown in Figure 14a,b. In the first configuration, the maximum displacements in the *X* and *Y* directions were as

follows: 0.0023 mm and 0.0027 mm, respectively, for 25 N; 0.0062 mm and 0.0171 mm, respectively, for 50 N. Then, in the second configuration, the results were as follows: 0.0217 mm and 0.0084 mm, respectively, for 25 N; 0.0289 mm and 0.0155, respectively mm for 50 N. Although the hysteresis loop obtained for the second orientation variant, shown in Figure 9b, has an irregular shape, the vibration trajectory shown in Figure 14b, obtained during the same series of measurements, has a regular shape. As noted earlier, the irregular shape of the hysteresis loop may occur due to fluctuations in the value of the measured force (this parameter has not been taken into account in Figure 14b).

The vibration trajectories presented in Figure 15 show that, at a frequency of 150 Hz, vibrations in the direction perpendicular to the direction of the acting load were predominant in most of the cases studied. The only exception was the case of the first configuration, where the system was loaded with a force of 50 N (Figure 15a). In this case, the maximum deviation of the sleeve from the equilibrium position was 0.0014 mm in the horizontal direction, and 0.0013 mm in the vertical direction. In the other three cases presented in Figure 15, the displacements of the sleeve in the vertical direction are several times higher than those in the horizontal direction. However, it is important to note that when the highest frequencies were tested (i.e., 150 Hz and 300 Hz), the movement of the bearing sleeve in the direction of the excitation force almost disappeared, and the vibration amplitude in this direction did not even reach a value of 1 µm. That is why a detailed analysis of the phenomena that accompany such small vibrations is of little practical significance. When considering the results of measurements presented in Figure 16, it can be further said that an increase in the excitation frequency to 300 Hz resulted in a reduction in the sleeve's vibration trajectory, which could be observed mainly in the vertical direction. Since the maximum deviations of the sleeve from the equilibrium position in the horizontal direction in this case also did not exceed 1 µm, such results were not analyzed in detail. It is also worth mentioning that, at high excitation frequencies, the position where the top foil was mounted relative to the direction of the acting load also had a strong effect on the shape of the characteristics obtained, and therefore, this can also have an influence on the dynamic characteristics of the foil bearing structure.

In order to sum up the tests carried out, a summary of the amplitudes of sleeve displacements obtained in all the measurement series has been presented in Figures 17 and 18. For ease of comparison, all graphs have the same displacement scale. Based on these results, it can be concluded that an increase in load from 25 N to 50 N in both configurations resulted in an increase in the vibration level, and the vibrations increased not only in the direction of the acting force (*X*) but also in the direction perpendicular to this direction (*Y*). It can also be noted that the increase in vibration was irregular, since in some cases, the ratios between the amplitude of the displacements in the *X* and *Y* directions inverted when the load increased. It can also be said that above the frequency of 80 Hz, these phenomena lost their importance because the vibration level was much lower.

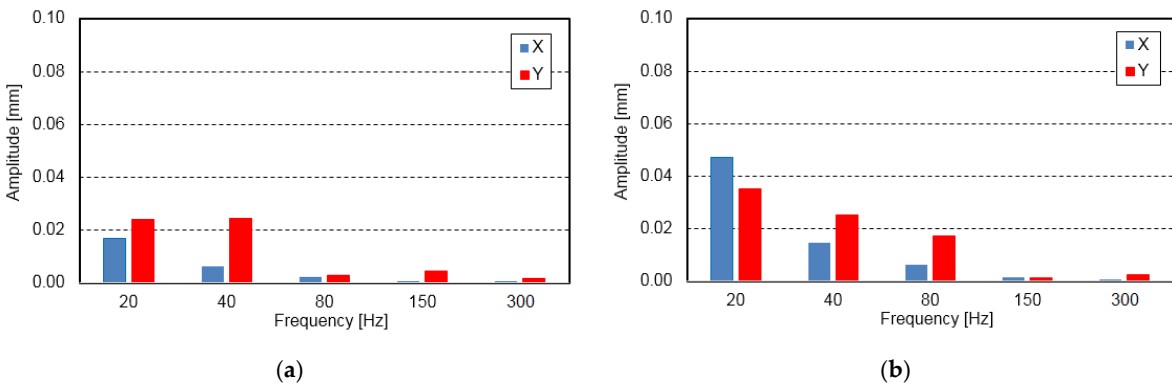

(**a**)                                                                     (**b**)

**Figure 17.** Summary of the amplitudes of sleeve displacements in two directions versus the frequency of the excitation force that acted in the direction perpendicular to the sleeve groove: (**a**) for a load of 25 N; (**b**) for a load of 50 N.

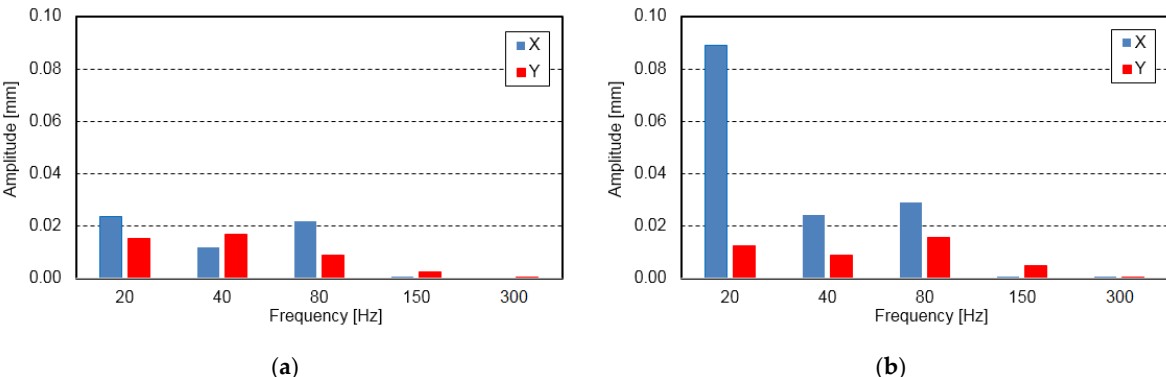

**Figure 18.** Summary of the amplitudes of sleeve displacements in two directions versus the frequency of the excitation force that acted in the direction parallel to the sleeve groove: (**a**) for a load of 25 N; (**b**) for a load of 50 N.

Since the results of the research carried out, which include all the experimental data used to present the hysteresis loops and vibration orbits in this article, could be useful to other researchers, the authors have decided to make these data available online as Supplementary Material for this article.

## 4. Conclusions

This article presents the results of an experimental study conducted on a foil journal bearing whose sliding surface of the top foil is coated with a polymer material. This study focused on evaluating the dynamic properties of the bearing structure, and the characteristics determined included hysteresis loops and orbits representing sleeve vibrations induced by dynamic loads. Based on the research carried out, the following conclusions can be drawn:

- Hysteresis loops have a very wide range of applications when it comes to analyzing the dynamic properties of a foil bearing structure. They can be used to evaluate the vibration damping capacity of the system. However, based on the study carried out, it is important to note that in the bump-type foil bearing structure, apart from the displacements that take place in the direction of the acting load, there are also displacements that occur in the perpendicular direction. These are due to the bearing design and the strong mechanical interactions that occur between the flexible components with complex geometries. Under certain load conditions, the displacements in the direction perpendicular to the direction of the acting force may be even greater than in the main direction, so they will also have a significant impact on the energy dissipation in such a system. Since the foil bearing structure has a high degree of anisotropy in terms of stiffness and damping, this must be taken into account when determining their stiffness and damping coefficients. This is a new finding that may change the approach used to determine the dynamic characteristics of a foil bearing structure. The sleeve displacements in the two directions must therefore be taken into account, both in the case of experimental methods and in the modeling and numerical analysis.
- It is important to note that the experimental studies discussed in this article concern a bump-type foil bearing with a specific design and materials, and an individually selected assembly preload. The results obtained may be adequate for a certain group of similar foil bearings, but should not be extended to all foil bearings. Since foil bearings are an evolving technology, and various modifications and improvements are being made all the time, which often have a very large impact on their characteristics, the authors of this article recommend an individual approach to each new design.
- The results of the study show that the anti-friction material used (polytetrafluoroethylene-based material with the trade designation AS20), with which one side of the top foil was coated, fulfilled its purpose perfectly. The run-in process of the foil bearing with

such a coating was very fast, and a temporary increase in the temperature of the top foil was followed by its stabilization at a very low level. After being run-in, the bearing exhibited low friction torque, and the protective coating was not damaged at all. After the target tests were carried out, during which the bearing structure was subjected to dynamic loads, the anti-friction coating was not damaged. The bearing was still suitable both for further dynamic tests and high-speed operations. The materials used in the tested bearing show great potential for use in real conditions.

The research discussed in this article will continue. The study of the influence of the assembly preload and that of the materials used for the anti-friction coating is also very important, not only from the point of view of the dynamic characteristics of the structure, but also from the point of view of the bearing as a whole. Indeed, preliminary studies carried out in this regard have shown that changing the bearing assembly preload and using materials with different tribological properties can have a significant impact on the dynamic characteristics of a foil bearing.

**Supplementary Materials:** The following supporting information can be downloaded at: https://www.mdpi.com/article/10.3390/coatings12091252/s1, Experimental data used to prepare Figures 7–16.

**Author Contributions:** Conceptualization, G.Ż.; methodology, G.Ż. and P.B.; investigation, P.B.; validation, G.Ż., P.B. and T.Z.K.; software, P.B., J.R. and P.Z.; formal analysis, P.B. and T.Z.K.; resources, P.B. and T.Z.K.; data curation, P.B., G.Ż., J.R.; writing—original draft preparation, G.Ż.; writing—review and editing, G.Ż. and A.M.; visualization, P.B., G.Ż., P.Z.; supervision, G.Ż. and A.M.; project administration, A.M. and G.Ż.; funding acquisition, A.M. and G.Ż.; All authors have read and agreed to the published version of the manuscript.

**Funding:** This research was funded by the National Science Center, Poland, grant number 2017/27/B/ST8/01822 entitled "Mechanisms of stability loss in high-speed foil bearings—Modeling and experimental validation of thermomechanical couplings".

**Institutional Review Board Statement:** Not applicable.

**Informed Consent Statement:** Not applicable.

**Data Availability Statement:** Not applicable.

**Conflicts of Interest:** The authors declare no conflict of interest.

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
