# Peer review of "Experimental Characterization of a Foil Journal Bearing Structure with an Anti-Friction Polymer Coating"

_coatings, doi:10.3390/coatings12091252_

Round 1

Reviewer 1 Report

Reviewer comments 

This paper stated on the “Experimental Characterization of a Foil Journal Bearing Structure with an Anti-Friction Polymer Coating”. This paper has been written well with high quality of scientific research. However, some parts should be revised before publication.

1-      The abstract was written well. However, it’s too long. The main highlight of your paper needs to be clearly showed.

2-      In the introduction section, the main purpose is not clearly explained. As well, the author should remove the unnecessary parts such as from line: 147-151, on page: 4.

3-      Please modify the image in Fig. 3. The dimensional view of gas foil bearing should be added.

4-      The conclusions are too long. Some sentences must be removed. Make them easier to understand.

Reviewer 2 Report

1.    The reference [7] is not cited in the text.

2.    More emphasis is required on the applied preload on the gas foil bearing. This will improve the understanding of the readers.

3.    The background of adopting technical dimensions for gas foil bearing as mentioned in Table 1 should be justified in the manuscript.

4.    Fig 6 (b), the irregular loop (shape change) in the 4th quadrant is reported. But possible reasons for the same are not presented in the manuscript.

5.     What is the basis of selecting the frequency viz. 20 Hz, 40 Hz, 80Hz, 150 Hz, and 300 Hz? And similarly, the load selection (25 N and 50 N) should be vindicated in the manuscript.

6.    In Fig 16, amplitude change does not follow the regular trend at 150Hz at 50N.  what are the reasons for the trend change in this particular condition?

7.    The conclusion section seems to be generic. At least some inferences must be quantified with their real implications.

Reviewer 3 Report

Here are the suggestions before it could be accepted.

1. The development of coating technology is very fast in recent years, the literature should be updated, at least 1/3 of the total literature should be in recent three years.

2. In the introduction, the disadvantages of the references should be summarized clearly to emphasize the importance of this work.

3. What is the accuracy of equipment in Fig.4 ? I suggest that a table is needed to describe all the equipment.

4. The conclusion should be simplified.  

Round 2

Reviewer 1 Report

The quality of paper is very improved. It can be accepted.